# The Vertical Metabolic Activity and Community Structure of Prokaryotes along Different Water Depths in the Kermadec and Diamantina Trenches

**DOI:** 10.3390/microorganisms12040708

**Published:** 2024-03-30

**Authors:** Hao Liu, Hongmei Jing

**Affiliations:** 1CAS Key Laboratory for Experimental Study under Deep-Sea Extreme Conditions, Institute of Deep-Sea Science and Engineering, Chinese Academy of Sciences, Sanya 572000, China; liuh@idsse.ac.cn; 2HKUST-CAS Sanya Joint Laboratory of Marine Science Research, Chinese Academy of Sciences, Sanya 572000, China; 3Southern Marine Science and Engineering Guangdong Laboratory (Zhuhai), Zhuhai 519000, China

**Keywords:** water column, benthic boundary layer, microbial metabolic activity, Kermadec Trench, Diamantina Trench

## Abstract

Prokaryotes play a key role in particulate organic matter’s decomposition and remineralization processes in the vertical scale of seawater, and prokaryotes contribute to more than 70% of the estimated remineralization. However, little is known about the microbial community and metabolic activity of the vertical distribution in the trenches. The composition and distribution of prokaryotes in the water columns and benthic boundary layers of the Kermadec Trench and the Diamantina Trench were investigated using high-throughput sequencing and quantitative PCR, together with the Biolog Ecoplate^TM^ microplates culture to analyze the microbial metabolic activity. Microbial communities in both trenches were dominated by Nitrososphaera and Halobacteria in archaea, and by Alphaproteobacteria and Gammaproteobacteria in bacteria, and the microbial community structure was significantly different between the water column and the benthic boundary layer. At the surface water, amino acids and polymers were used preferentially; at the benthic boundary layers, amino acids and amines were used preferentially. Cooperative relationships among different microbial groups and their carbon utilization capabilities could help to make better use of various carbon sources along the water depths, reflected by the predominantly positive relationships based on the co-occurrence network analysis. In addition, the distinct microbial metabolic activity detected at 800 m, which was the lower boundary of the twilight zone, had the lowest salinity and might have had higher proportions of refractory carbon sources than the shallower water depths and benthic boundary layers. This study reflected the initial preference of the carbon source by the natural microbes in the vertical scale of different trenches and should be complemented with stable isotopic tracing experiments in future studies to enhance the understanding of the complex carbon utilization pathways along the vertical scale by prokaryotes among different trenches.

## 1. Introduction

Prokaryotes play a crucial role in remineralizing and transforming significant amounts of different types of organic matter that exist in seawater, including suspended sediment particles, phytoplankton debris, living plankton, zooplankton fecal materials, aggregates, marine snow, transparent polymeric particles, colloidal particles, and so on [1,2]. This microbially transformed carbon may influence carbon export efficiency by facilitating aggregation/disaggregation activities, depolymerization and degradation [3,4]. Most particulate organic matter (POM) generated from the euphotic zone is degraded and remineralized while sinking through the water column (WC), leading to about 1–40% organic matter transported vertically from the surface ocean to the deep ocean [5,6]. Although the majority of organic matter is consumed in the photic zone, the fate of the sinking particles that reach the deep sea are largely determined by the abundance, diversity, and metabolic activity of microbial communities along the water depth profile [5].

The physicochemical conditions in the ocean WC are not uniform, but varied with increasing depths [7], providing strong selective pressures on microbial communities along the vertical WC [8,9]. These deep-sea communities can differ from the waters above and showed different microbial diversity and metabolic rates [10]. These variations may reflect carbon source availability [11] and dispersal limitation [12]. In addition, the benthic boundary layer (BBL), defined as the bottom layer of the water column directly adjacent to the seabed [13], contains a high concentration of particles resuspended from subsurface sediments [14]. The carbon sources and compositions in the BBL were reported to be distinctly different from the sinking POM and, thus, might affect the diversity and metabolic function of the microbial community. Consequently, molecular ecological studies on microbes from the euphotic layer to the BBL would contribute to understanding the microbially driven organic carbon transformations, as well as linking specific microbial groups involved in the relevant metabolic processes in different water layers.

The Biolog EcoPlates^TM^ method is a simple and sensitive way to reveal the functional diversity of microbial communities by relying on their microbial capability to use various carbon sources. It has been used to assess the metabolic activity of microbes in different water depths of the South China Sea [15] and surface sediments of the Mariana Trench [16]. But there is still a lack of direct evidence of carbon source utilization by microbial groups from the euphotic zone to abyssal depths. Carbon sources and compositions of organic carbon have been reported to vary with different water depths [17]; therefore, culture-dependent EcoPlates cultivation together with culture-independent molecular study would help to obtain a quick glimpse of the shift in the ecological roles of microbial communities from the euphotic layer to the BBL.

Hadal trenches are unique deep-sea environments with distinct benthic communities, due to their steep topography and periodic disturbance by turbidity flows [18]. The Kermadec Trench is located about 120 km off the northeastern coast of New Zealand in the Southern Hemisphere. It reaches a maximum depth of 10,047 m, making it the fifth deepest trench [19]. It is 1500 km long, with a mean width of 60 km and exhibits the characteristic V-shape cross section common to hadal trenches. The Diamantina Trench is located in the Indian Ocean and has a maximum depth of approximately 8047 m. It is approximately 520 km long and 70 km wide, running in a northeast–southwest direction. It is around 1500 km west of Perth, Australia. As part of the broader coordinated effort to explore the biogeochemistry and ecology of different carbon sources along the vertical WC in hadal trenches, prokaryotes were collected from five water depths, together with four samples from the BBL in the Kermadec and Diamantina Trenches, respectively. The diversity and composition of the microbial communities were studied with high-throughput sequencing and quantitative PCR (qPCR). The Biolog EcoPlates^TM^ method was also applied to investigate the carbon metabolic capability of microbes. This comparative study will contribute to a better understanding of the shifts and connectivity in the diversity and specific carbon metabolic capabilities of microbial communities from the surface seawater to the BBL in the Kermadec and Diamantina Trenches.

## 2. Materials and Methods

### 2.1. Sample Collection and Environmental Factor Measurement

Seawater samples were collected from the Kermadec and Diamantina Trenches during cruise TS29 from November 2022 to March 2023 (Figure 1). Niskin bottles (General Oceanics, Miami, FL, USA) were used to collect WC samples from five different depths (i.e., 0 m, 200 m, 800 m, 2000 m, and 5000 m). BBL samples were collected by the R/V “Fen Dou Zhe” from another eight stations with water depths ranging from 2311 to 9639 m (Table 1). Approximately 2 L of seawater was filtered through 200 μm mesh and was then sequentially filtered through the 3 and 0.22 μm pore size polycarbonate filter (47 mm, EMD Millipore, Billerica, MA, USA). All the filters were then flash frozen and stored at −80 °C until DNA extraction in the laboratory. The in situ environmental parameters (temperature, salinity, and depth) were recorded with a conductivity–temperature–depth (CTD, Sea-Bird Electronics, Bellevue, WA, USA) and the CTD (SBE-49, Seabird Electronics) in the manned submersible the R/V “Fen Dou Zhe”. The concentrations of inorganic nutrients (NO_3_^−^ + NO_2_^−^, NH_4_^+^ and PO_4_^3−^) were analyzed using an auto-analyzer (QuAAtro, Blue Tech Co., Ltd., Tokyo, Japan).

### 2.2. Microbial Metabolic Activity Analyses

Water samples were filtered through 3 μm pore size polycarbonate filters, and approximately 150 μL of each filtrate was added to each well of the Biolog EcoplateTM microplates. The Ecoplates for samples collected from 0 m were incubated at room temperature (~25 °C), those from 200 m were incubated at room temperature without light (one layer of black plastic bag), and those from the other depths were incubated at ~4 °C without light (two layers of black plastic bag). The absorbance of each well was measured at 590 nm and 750 nm every 24 h for a total of 100 days of continuous cultivation. The difference in absorbance value was used to characterize the color change of the Biolog Ecoplate^TM^ microplates by eliminating the change in turbidity caused by fungi at 750 nm. The average well color development (AWCD) was calculated to determine the utilization of carbon sources and metabolism characteristics.
(1)$$\mathrm{AW}CD=\frac{\sum R-C}{n}$$

In Equation (1), *R* is the absorbance of each well, *C* is the absorbance of the control well, and *n* is the number of substrates present in the particular category [20]. Meanwhile, the richness (R), Simpson (D), and Shannon (H) indices were calculated to reflect the functional diversity of the microbial community. The richness index refers to the number of oxidized substrates. It was calculated as the sum of the OD_i_ value of cells which were at least 0.5 after incubation [21], where OD_i_ is the corrected OD value of each individual well, in two consecutive measurements at two different measurement times for tn and t_n + 1_.

The Simpson index (D) was calculated using the following Equation (2):(2)$$D=-\ln\sum_{i=1}^{N}{\left(Pi\right)}^{2}$$
where *N* is the number of substrates and *P_i_* is the relative color development of the well over the total color development of each well of a plate [22].
(3)$$\mathrm{Pi}=\frac{\mathrm{ODi}}{\sum_{i=1}^{N}ODi}$$

The Shannon index was calculated according to the following Equation (4):(4)$$H\;=-\sum_{i=1}^{N}Pi.\ln Pi$$
where *P_i_* and *N* are the same as in the calculation of the Simpson index [23].

### 2.3. DNA Extraction, PCR Amplification, and Sequencing

Genomic DNA was extracted from 0.22 μm pore sized polycarbonate filters using a PureLink Genomic DNA Mini Kit (Invitrogen, Thermo Fisher Scientific, Corp., Carlsbad, CA, USA). The DNA was amplified using PCR with universal prokaryotic primers of Pro341F (5′-CCTACGGGNBGCASCAG-3′) and Pro805R (5′-GACTACNVGGGTATCTAATCC-3′) [24], which target the V3–V4 region of the 16S rRNA gene. The primers were tagged with a 6 bp barcode to distinguish amplicons in the pools of all samples multiplexed for Illumina sequencing. PCR amplification was performed in triplicate on a BIO-RAD C1000 Touch^TM^ Thermal Cycler PCR System in a 20 μL PCR reaction mix, containing 2.0 μL 10 × PCR-MgCl_2_ buffer, 0.7 μL 2.5 mM dNTPs, 0.7 μL MgCl_2_, 0.8 μL forward primer, 0.8 μL reverse primer, 0.2 μL Platinum^®^ TaqDNA polymerase, 2.5 μL template DNA, and 12.3 μL ddH_2_O. Thermal cycling was performed at 95 °C for 3 min, followed by 33 cycles at 95 °C for 0.5 min, 55 °C for 45 s, 72 °C for 30 s, and a final extension at 72 °C for 8 min. Double-distilled water was used as a negative control. Amplification and paired-end sequencing of the amplicons were then performed with an Illumina HiSeq PE250 sequencer (Novogene Co., Ltd., Beijing, China, www.novogene.com, accessed on 5 March 2024).

### 2.4. Quantitative PCR

The abundance of the 16S rRNA gene was quantified using real-time quantitative PCR via a StepOnePlus Real-Time PCR system (Applied Biosystems Inc. Carlsbad, CA, USA). Each qPCR reaction comprised 10 μL 2 × SYBR^®^ Premix Ex Taq^TM^II (Takara Bio. Inc., Shiga, Japan), 0.3 μM Uni340F (5′-CCTACGGGRBGCASCAG-3′)/Uni806R (5′-GGACTACNNGGGTATCTAAT-3′) primers [25], 2 μL DNA as the template, 0.4 μL ROX reference dye, and water to a total of 20 μL. Quantitative PCR reactions and calibrations were performed as previously reported [25]. Triplicate qPCR reactions were performed for each sample with an efficiency of ~101.7% and the gene copy number was normalized to gene abundance.

### 2.5. Bioinformatics Analysis

After sequencing, barcoded and low-quality sequences were removed using QIIME 2 with default parameters [26]. Chimeras were detected and removed using UCHIME against the SILVA database release 128 [27] and reads presented as a single copy (i.e., singletons) were manually removed. The remaining reads were then clustered into the Amplicon Sequences Variant (ASV) using DADA2 (Divisive Amplicon Denoising Algorithm) [28]. Taxonomy assignment of ASVs that were not affiliated with prokaryotes, as determined from the SILVA database release 138, were further removed [26]. A filtered ASVs table was generated for each sample with QIIME 2, and the Shannon diversity index and Evenness was calculated. The microbial community structures were visualized by stacked plot using the “ggplot2” package in R (version 3.5.3). Network analysis was conducted to explore the co-occurrence patterns within/between the taxa of prokaryotes. A similarity matrix was first generated by inputting a typical ASV matrix file and then the correlation matrix, r value, and *p* value were calculated using corr. test in the “psych” package [29] of R version 3.5.3. ASVs, which are strongly and significantly correlated (Spearman’s |r| > 0.6 and FDR-adjusted *p* < 0.05), were used to construct the networks using Gephi version 0.9.3 [30]. In addition, predicted potential carbon, nitrogen, and sulfur cycle-related pathways based on the 16S rRNA gene were conducted with the open-source R package Tax4Fun [31], with the short reads mode disabled along with SILVA 16S rRNA database (version 138), and were visualized by bubble plots using the “ggplot” package.

### 2.6. Statistical Analysis and Accession Number

The non-metric multidimensional scaling (nMDS), based on the Bray–Curtis similarity index, was calculated using Paleontological Statistics (PAST) version 3 [32]. An analysis of similarities (ANOSIM), based on the ASVs’ relative abundance was conducted with Paleontological Statistics (PAST) version 3 [32] to test whether there was a significant difference in the microbial community structure and potential metabolic function among the different samples. Values of *p* < 0.05 and *p* < 0.01 were considered to indicate different levels of statistical significance. The Statistical Analysis of Metagenomic Profiles (STAMP, version 2.1.3) was performed according to the relative abundance of each ASV [33] and was visualized using R (version 3.5.3). *p* values were calculated using a two-sided Welch’s *t*-test with the Benjamin–Hochberg False Discovery Rate (FDR) correction. A redundancy analysis (RDA) was performed with CANOCO V5.0 to identify a possible differentiation of the communities under the constraint of environmental factors. The statistical significance of an explanatory variable added in the course of forward selection was tested with the Monte Carlo permutation test (9999 permutations, *p* < 0.05). The phylogenetic group data were Hellinger transformed, environmental variables were logarithm transformed, and the effects of collinearity (VIF > 10) were removed.

16S rRNA gene sequences obtained from this study have been deposited in the National Center for Biotechnology Information (NCBI) Sequence Read Archive (SRA), under the accession number PRJNA1059802.

## 3. Results

### 3.1. Hydrographic Conditions of Sampling Stations

In both trenches, seawater temperatures decreased sharply from 19.50 °C~20.24 °C in the surface to 1.07 °C~1.12 °C at the depth of 5000 m along the water column, while the average temperatures in the BBL ranged between ~1.79 °C and 1.36 °C, respectively (Figure 2). Salinity was highest for the surface water and decreased with depths to 800 m, and then slightly increased with depth to 5000 m in the WC in the two trenches, while it was generally the same in the BBL of the two trenches. Generally, the concentrations of ambient inorganic nutrients were always higher in the Diamantina Trench than those in the Kermadec Trench, except for NH_4_^+^ in the WC. The concentrations of PO_4_^3−^ increased with depth and reached a maximum at 800 m, then decreased with depths in the Kermadec Trench, while those in the Diamantina Trench increased with depth and reached a maximum at 5000 m, except for 2000 m. The NO_3_^−^ + NO_2_^−^ concentrations in the WC were both increased with depth in both trenches. The NH_4_^+^ concentrations peaked at 0 m in the Kermadec Trench and at 5000 m in the Diamantina Trench and the minimum values were detected at 2000 m in both trenches (Figure 2A,B). All the ambient inorganic nutrients were significantly higher in the BBL than in the WC (*p* < 0.05, Figure 2).

### 3.2. Microbial Community Composition and Diversity

In the Kermadec Trench, 279,116 sequences and 2095 ASVs were generated from the microbial community, with the highest and lowest numbers of ASVs found at 2000 m and Stn. FDZ152, respectively (Table 1). In the Diamantina Trench, 342,737 sequences and 1443 ASVs were generated from the microbial community, with the highest and lowest numbers of ASVs found at 0 m and Stns. FDZ172 and FDZ175, respectively (Table 1). In the Kermadec Trench, the microbial community structure was dominated by Thermoplasmata, Nitrososphaeria, and Halobacteria in archaea, and by Alphaproteobacteria, Gammaproteobacteria, and Actinobacteria in bacteria. The community structures of the euphotic layer were significantly different from those of other layers (ANOSIM, *p* < 0.05), while those among different BBLs had no significant difference (Figure 3A). In the Diamantina Trench, the microbial community structure was dominated by Nitrososphaeria and Halobacteria in archaea, and Alphaproteobacteria and Gammaproteobacteria in bacteria. The community structure of the 2000 m layer was significantly different from that of the other water layers (ANOSIM, *p* < 0.05, Figure 3B), while those among different BBLs were basically the same. The lowest Shannon index and Evenness were present at 2000 m in both trenches. The microbial community diversity between WC and BBL was only significantly different in the Kermadec Trench (*p* < 0.05, Figure 3C), but was not observed in the Diamantina Trench (Figure 3D). An NMDS plot demonstrated a distinct distribution of microbial assemblages between the WC and BBL in both trenches (ANOSIM, *p* < 0.05, Figure 3E,F). Based on the STAMP analysis, the indicative ASVs leading to significant differences between the WC and BBL in the Kermadec Trench were ASV91 (*Alteromonadales*, Gammaproteobacteria) and ASV825 (*Cellvibrionales*, Gammaproteobacteria) (*p* < 0.05, Appendix A), while the indicative ASVs leading to significant differences between the WC and BBL in the Diamantina Trench were ASV449 (*Propionibacteriales*, Actinobacteria) and ASV825 (*Cellvibrionales*, Gammaproteobacteria) (*p* < 0.05, Appendix A).

### 3.3. Gene Abundance and Environmental Impacts

For microbial gene abundance based on qPCR, it was significantly higher at the BBL than the water columns in both trenches (*p* < 0.05), and that in the BBL was significantly higher in the Diamantina Trench than in the Kermadec Trench (*p* < 0.05, Figure 4A,B). RDA analysis revealed that the first and second axes together contributed 95.05% and 98.56% to the total variance of the whole communities in the Kermadec and the Diamantina Trenches, respectively (Figure 4C,D). Among the tested environmental factors, temperature (*p* < 0.05), depth (*p* < 0.05), and NO_3_^−^ + NO_2_^−^ (*p* < 0.01) were statistically significant, contributing to the variation of prokaryotic communities in the Kermadec Trench (Figure 4C), but there was no significant environmental factor detected in the Diamantina Trench (Figure 4D).

### 3.4. Co-Occurrence Network and Potential Metabolic Functions of Prokaryotes

To elucidate the interactions between different microbial groups in the WC and the BBL, network analyses were conducted based on the top 200 ASVs (Figure 5). The modularity index was 0.94 and 0.81 of the WC and the BBL in the Kermadec Trench, while it was 0.92 and 0.68 of the WC and the BBL in the Diamantina Trench. The index was larger than 0.4, suggesting that the network had a modular structure (Newman, 2006). Mainly positive correlations were shown among the archaeal groups, while more negative correlations were found among different bacterial groups. The correlations between different groups of archaea and bacteria were mainly positive, with the proportions of positive correlations accounting for 84.4% and 77.5% of the WC and the BBL in the Kermadec Trench (Figure 5A,B), respectively; and accounting for 90.4% and 95.4% of the WC and the BBL in the Diamantina Trench, respectively (Figure 5C,D). The detected negative correlations were mainly between Proteobacteria and Cyanobacteria of the WC and Proteobacteria and Actinobacterio of the BBL in the Kermadec Trench (Figure 5A,B), and between Proteobacteria, Bacteroidota, and Actinobacterio of the WC and Proteobacteria and Actinobacterio of the BBL in the Diamantina Trench (Figure 5C,D).

Potential functions related to carbon, nitrogen, and sulfur metabolism were predicted based on the 16S rRNA genes by Tax4Fun (Appendix A). Generally, the abundance of carbon metabolism was higher than the nitrogen and sulfur metabolism. Hydrocarbon degradation and aerobic chemoheterotrophy were the major carbon metabolic processes in the two trenches. The functions related to dark oxidation of sulfur compounds and sulfide/sulfite oxidation metabolism were only found in the BBL and were significantly higher in the Diamantina Trench than in the Kermadec Trench (*p* < 0.05).

### 3.5. Prokaryotic Metabolic Activity

Biolog Ecoplate^TM^ microplates contained 31 different carbon sources belonging to 6 carbon categories, which were carbohydrates, polymers, phenolic, carboxylic, amino acids, and amines. The average well color development (AWCD) was calculated to determine the utilization of carbon sources and metabolism characteristics. Based on the AWCD, compared to the Kermadec Trench, carbon source utilization was more consistent across the Diamantina Trench, except for at 800 m. The utilization of different carbon categories by microbes was significantly higher at 0 m in the WC of the Kermadec Trench (*p* < 0.01) (Figure 6A). Metabolic diversity (AWCD, Richness, Simpson, and Shannon) was significantly lower at 800 m than at the other depths in the WC of the two trenches (*p* < 0.05; Figure 6). Metabolic activity reflected by the AWCD demonstrated that microbes first entered the exponential growth period and then the stable period during cultivation. The microbial community at 0 m entered the stable period (34 days) prior to other samples (40 days) in the Kermadec Trench. Meanwhile, microbial communities at 0 m and 800 m depths entered the stable period (20 days) prior to other samples (40 days) in the Diamantina Trench. In the surface water, amino acids and polymers were preferentially used; in the benthic boundary layers, amino acids and amines were preferentially used in the two trenches (Figure 7). The utilization of carboxylic acids was significantly higher in the BBL of Stns. FDZ154 and FDZ156 in the Kermadec Trench (*p* < 0.01, Figure 7G). In the Diamantina Trench, the utilization of carboxylic acids was significantly higher in samples taken from 0 m and 200 m, compared to the other samples (*p* < 0.01, Figure 7H). The utilization of phenolic acids was significantly higher in the BBL of Stns. FDZ154 and FDZ156 of the Kermadec Trench (*p* < 0.01, Figure 7G) than other samples. In the Diamantina Trench, phenolic acids were the least used and there were no significant differences among all the depths (Figure 7H). The utilization of amines was significantly higher in the BBL than in the WC, except at 0 m in the Kermadec Trench (*p* < 0.05, Figure 7K), while no significant differences between the WC and the BBL were observed in the Diamantina Trench (Figure 7L).

## 4. Discussion

### 4.1. Geographical Distribution and Environmental Effects

In the two trenches, the significantly different community structures between the WC and the BBL might be attributed to the in situ concentration and availability of organic matter. Generally, the BBL contains a high concentration of particles resuspended from subsurface sediments [14,34], which could explain the significantly higher gene abundance of prokaryotes in the BBL than in the WC. A higher proportion of Gammaproteobacteria in the BBL was found in the Kermadec Trench than in the Diamantina Trench and this microbial group could attach to particles to avoid the nutrient-depleted conditions in the surrounding waters [35] and easily assimilated organic carbon sources [36]. Proteobacteria and Nitrososphaeria predominate in the WC and this was generally consistent with previous studies conducted in the Kermadec Trench [37]. Halobacteria as a class of Euryarchaeota are extremely halophilic archaea that can adapt to a wide range of salt concentrations [38] and could catalyze the terminal step in the degradation of organic matter in anoxic environments where light was limiting [39]. The community structure of the euphotic layer was distinct from that of other layers in the Kermadec Trench, this might be due to the higher relative proportions of Cyanobacteria, which play an important role in the uptake and conversion of CO_2_ to bioproducts through their photosynthetic system [40].

The most important impacting factors in the Kermadec Trench were the sampling depth, temperature, and NO_3_^−^ + NO_2_^−^ concentrations. Sampling depth has been reported as a significant driver of community composition in the Mariana Trenches [37]. Temperature could affect the community composition and metabolic activity involved in the remineralization of POC in the WC [15,41]. In addition, NO_3_^−^ was reported as the primary chemical parameter affecting the microbial community composition [42]. While no significant impacting factors were identified for microbial communities in the Diamantina Trench, this was possibly due to other important factors that were not measured, such as dissolved organic/inorganic carbon, POC, and that should be included in future studies.

### 4.2. Carbon Source Utilization and Potential Metabolic Activity

Heterotrophic microorganisms are crucial in the decomposition of POM and the subsequent remineralization processes in the seawater. Although natural carbon sources should be more complex than those contained in the Biolog Ecoplate^TM^ microplates, substrates which were promptly and preferentially utilized during the microplate incubation might provide important information on the carbon source in natural environments [43]. At the surface water, amino acids and polymers were used preferentially; at the BBL, amino acids and amines were used preferentially. The latter might be attributed to the presence of a high proportion of Alphaproteobacteria. This bacterial group can utilize a range of organic compounds, mainly including amino acids, nucleic acids, fatty acids, and other low molecular weight compounds, including organic and aromatic hydrocarbons produced by algae [44,45]. Amino acids were preferentially used in both the WC and BBL in the two trenches, which could be due to the fact that they were most used in cooperative communities, which was found in previous studies [46,47]. In this study, the microbial metabolic activity, reflected by the AWCD values, at 800 m was significantly different from those at the other depths. The enzyme active genes of carbohydrate esterases and auxiliary activities was reported to be positively related to the salinity [48] and this might explain the lowest microbial metabolic activities at 800 m, which always showed the lowest salinity in the WC in both trenches.

Microorganisms commonly metabolize simple carbohydrates, such as glucose, due to their easy digestibility and prevalence [49]. Phytoplankton exudates were the main source of dissolved carbohydrates in marine systems [50] and this might explain the higher metabolic activity involving carbohydrates in the surface layer of the Kermadec Trench, where a higher relative abundance of Cyanobacteria is harbored. Amino acids were primarily derived from marine biodegradation, protein hydrolysis, and extracellular excretion and were important components of marine organic nitrogen and organic carbon. Some amines, such as putrescine, are produced during the degradation of amino acids. The degradation of complex polymers requires a complex metabolic capability and the biogenic amines which are produced by bacteria were controlled by the high hydrostatic pressure and low temperature [51,52]. In addition, annotated functional genes could provide an assessment of microbial metabolic activities, allowing for the elucidation of various processes associated with substance and energy metabolism. In this study, the high abundance of aerobic chemoheterotrophy in the BBL were significantly different from those in the WC (*p* < 0.05) in the Diamantina Trench. This might be due to the fact that the BBL contained a high concentration of resuspended particles from subsurface sediments as extra carbon sources, and heterotrophic prokaryotes transform the organic matter to obtain energy and carbon substances via an aerobic process [53].

### 4.3. Microbial Interaction and Ecological Significance

Microbial interactions could lead to a series of competitive or collaborative relationships and have been suggested as biotic drivers that affect microbial community composition [54,55]. More positive relationships were detected in the WC and BBL of the two trenches, indicating the possibility of cross-feeding, co-colonization, and niche overlap [56]. Different microbial groups and their carbon utilization capabilities could help to make better use of various carbon sources along the water depths. A previous study showed that the heterotrophic archaea preferentially used heavy carbon sources, such as algal carbohydrates [57], and Crenarchaeota had genes for using carbohydrates as an organic carbon source and genes for transporting amino acids from the environment [58]. Proteobacteria contain a diverse functional repertoire including their chemolithotrophic ability to utilize sulfur and C1 compounds and their chemo-organotrophic ability to utilize environment-derived fatty acids, aromatics, carbohydrates, and peptides [59]. Some groups of Proteobacteria could rapidly utilize D-glucose and produced refractory dissolved organic matter that persisted for more than a year [60]. These organic moieties from resistant polymers could be utilized by deep-sea microorganisms due to their high ectoenzymatic activity [61,62]. For example, Chloroflexi harbored pathways for the complete hydrolytic or oxidative degradation of various recalcitrant organic matters [15], and Bacteroidota were possibly responsible for the decomposition of polymers [63]. In addition, Actinobacteria have evolved with numerous biosynthetic gene clusters to produce diverse bioactive secondary metabolites [64]. Those microbial groups were very likely worked together with diverse organic matter accessing strategies, contributing to the degradation of marine organic matter in the WC [65].

The two trenches exhibited different fluxes of organic matter. The annual rates of primary production in the overlying waters of the Kermadec Trench, which was the first trench in the Pacific Ocean to receive Lower Circumpolar Deep Water [66], have been estimated as 87 g C m^−2^ yr^−1^. In contrast, a higher surface chlorophyll in the euphotic layer of the Diamantina Trench of the Indian Ocean [67,68] might affect the sinking carbon sources and microbial community composition as well [69]. Significant differences in microbial metabolic activity (AWCD) existed between the WC and BBL in the Kermadec Trench, consistent with the NMDS plot. This might be due to the vertical shifts of community composition leading to the utilization shift of the organic detritus, which might be due to different accessing strategies of organic matter by heterogeneous microorganisms [65,69]. The microbial metabolic activity (AWCD) at 800 m was significantly lower from those at the other depths found in both trenches (*p* < 0.05), which was also found in the South China Sea [15]. The water depth at 800 m is usually defined as the lower boundary of the twilight zone [70] and the majority of sinking organic matter in the WC was in the form of marine snow, fecal pellets, and other particles of detritus and might have been previously ingested and reworked multiple times by zooplankton with selective absorbance of the most labile and nutritious dietary compounds above this layer [71], leading to an increase in the relative proportions of refractory polysaccharides in detritus at this depth. It should be noted that the microplate used in this study had limited types of carbon sources and the incubation results only provide a quick look at the carbon source preference and might not accurately reflect the actual carbon source compositions found in natural seawater. Therefore, the mineralization rate and the biochemical fate of different carbon sources into inorganic carbon via respiration and microbial assimilation at different depths needs to be further studied using stable isotopic tracing experiments.

## Figures and Tables

**Figure 1 microorganisms-12-00708-f001:**
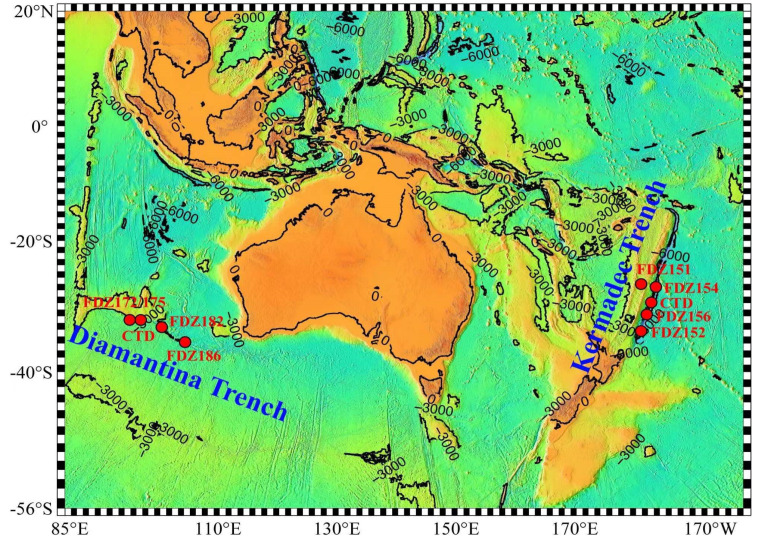
Map of the sampling stations used in this study.

**Figure 2 microorganisms-12-00708-f002:**
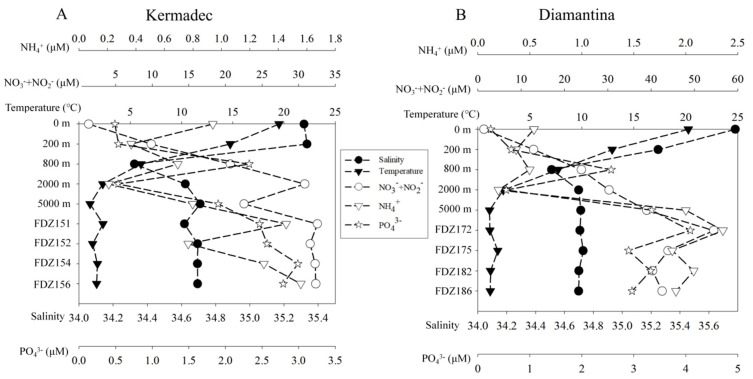
The temperature, salinity, and inorganic nutrients of ambient water samples of the Kermadec Trench (**A**) and the Diamantina Trench (**B**). The black circle and triangle symbols stand for salinity and temperature, while the blank circle, triangle, and star symbols stand for NO_2_^−^ + NO_3_^−^, NH_4_^+^, and PO_4_^3−^, respectively.

**Figure 3 microorganisms-12-00708-f003:**
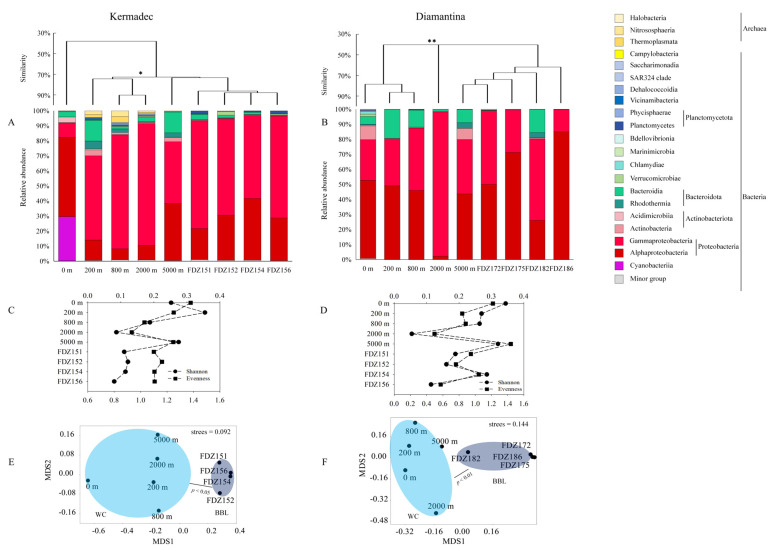
Microbial community structure in the WC and BBL of the Kermadec Trench (**A**) and the Diamantina Trench (**B**). * and ** stand for *p* < 0.05 and *p* < 0.01. The Shannon index and Evenness index in the WC and BBL of the Kermadec Trench (**C**) and the Diamantina Trench (**D**). The black circle and square symbols stand for the Shannon index and Evenness index, respectively. Non-linear multidimensional scaling (nMDS) analysis of microbial communities in the WC and BBL of the Kermadec Trench (**E**) and the Diamantina Trench (**F**) based on Bray–Curtis distances.

**Figure 4 microorganisms-12-00708-f004:**
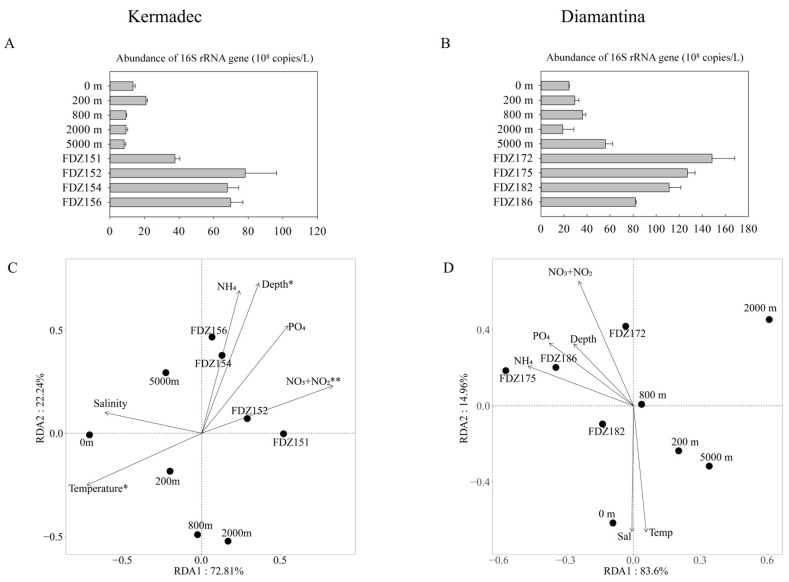
The abundance of the 16S rRNA gene at the WC and BBL of the Kermadec Trench (**A**) and the Diamantina Trench (**B**). Biplot of the redundancy analysis integrating environmental parameters and the microbial communities in the WC and BBL of the Kermadec Trench (**C**) and the Diamantina Trench (**D**). * and ** stand for *p* < 0.05 and *p* < 0.01.

**Figure 5 microorganisms-12-00708-f005:**
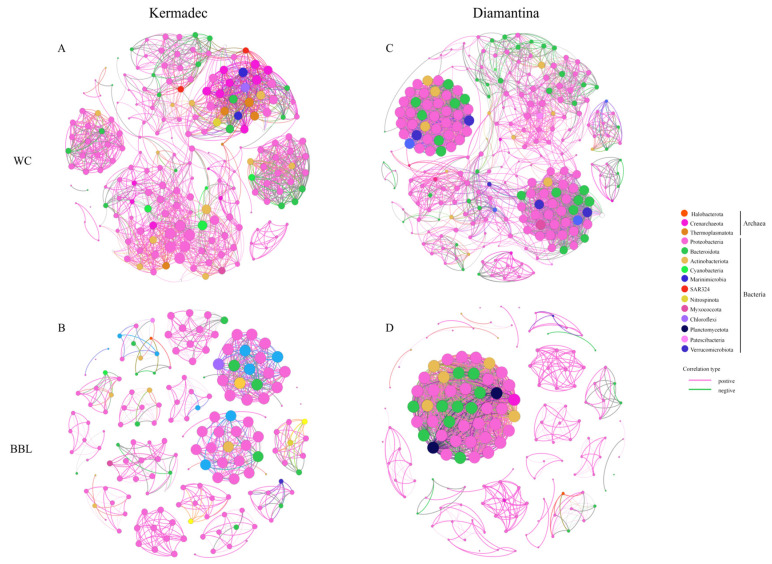
The networks analysis among archaeal and bacterial groups in the WC and BBL of the Kermadec Trench (**A**,**B**) and the Diamantina Trench (**C**,**D**). The network represents relationships between co-occurring ecosystems, the edges represent co-occurrence relationships consistent at the 0.6 correlation level, and the nodes represent archaeal and bacterial taxa.

**Figure 6 microorganisms-12-00708-f006:**
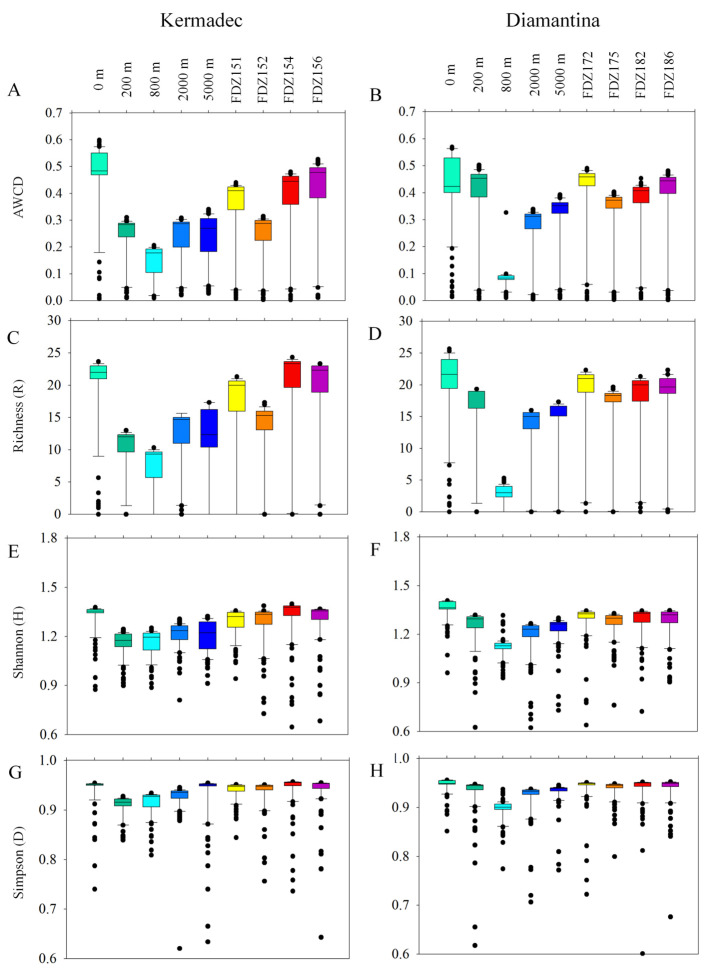
Box plots of the microbial metabolic activity of AWCD (**A**,**B**), Richness index (**C**,**D**), Shannon index (**E**,**F**), and Simpson index (**G**,**H**) in the WC and BBL of the Kermadec Trench and the Diamantina Trench.

**Figure 7 microorganisms-12-00708-f007:**
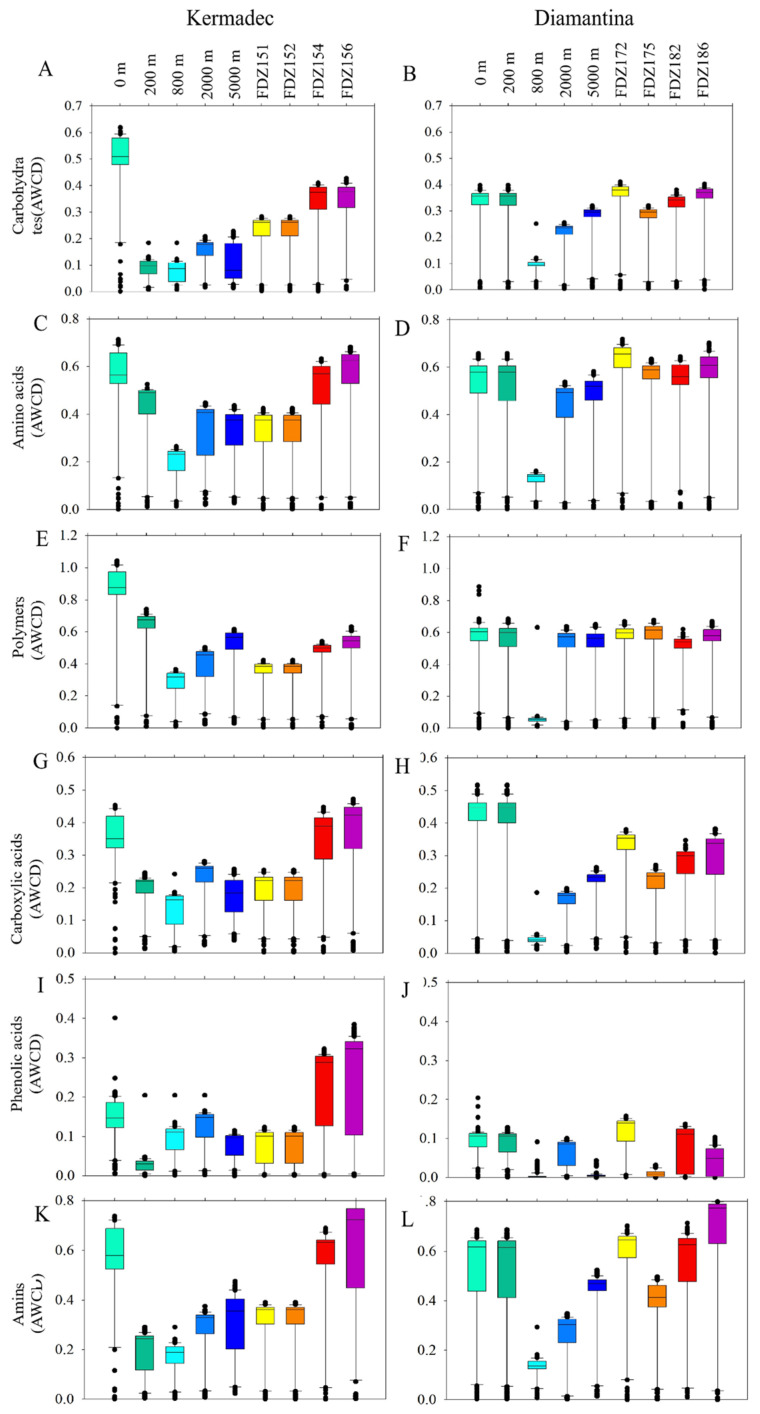
Box plots of the utilization capability of the six major carbon groups, carbohydrates (**A**,**B**), amino acids (**C**,**D**), polymers (**E**,**F**), carboxylic acids (**G**,**H**), phenolic acids (**I**,**J**), and amines (**K**,**L**) by microbes in the WC and the BBL of the Kermadec Trench (left panel) and Diamantina Trench (right panel).

**Table 1 microorganisms-12-00708-t001:** Sequencing information and diversity parameters of the 16S rRNA gene in this study.

Regions	Stns.	Longitude	Latitude	Depth (m)	Original Reads	Quality Reads	ASVs
Kermadec Trench	CTD-0 m	−176.4300	−29.9167	0	76,240	23,118	207
	CTD-200 m	−176.4300	−29.9167	200	63,214	28,839	318
	CTD-800 m	−176.4300	−29.9167	800	66,628	23,559	316
	CTD-2000 m	−176.4300	−29.9167	2000	81,419	25,578	371
	CTD-5000 m	−176.4300	−29.9167	5000	61,903	33,156	176
	FDZ151	−178.1515	−27.0460	2311	67,341	39,850	194
	FDZ152	−178.1519	−34.2276	6969	62,866	34,872	158
	FDZ154	−175.6629	−27.4976	9639	63,346	36,404	193
	FDZ156	−177.1942	−31.7540	9165	62,593	33,740	162
Diamantina Trench	CTD-0 m	97.7227	−32.5819	0	75,374	33,225	278
	CTD-200 m	97.7227	−32.5819	200	78,964	34,370	195
	CTD-800 m	97.7227	−32.5819	800	79,762	39,750	166
	CTD-2000 m	97.7227	−32.5819	2000	80,092	28,715	125
	CTD-5000 m	97.7227	−32.5819	5000	74,029	32,712	173
	FDZ172	95.8837	−32.5689	5164	68,712	40,635	114
	FDZ175	95.8820	−32.5679	3060	71,551	46,621	114
	FDZ182	101.2575	−33.6463	6900	66,139	37,606	163
	FDZ186	105.1887	−35.8289	6594	76,933	49,103	115

## Data Availability

16S rRNA gene sequences obtained from this study have been deposited in the National Center for Biotechnology Information (NCBI) Sequence Read Archive (SRA) under the accession number PRJNA1059802.

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
