# Peer review of "The Vertical Metabolic Activity and Community Structure of Prokaryotes along Different Water Depths in the Kermadec and Diamantina Trenches"

_microorganisms, 2024, doi:10.3390/microorganisms12040708_

Round 1

Reviewer 1 Report

Comments and Suggestions for Authors

1. You must be insert a map area.

2. Why didn't you measure the redox potential?? For this purpose, given the "extreme" environmetal conditions, it would have been important to correlate the structure of the microbial community to the different concentrations of oxygen at various depths. You have postulated a correlation between the structure of the microbial community and variation in environmental parameters, if you have measured the redox, but have not reported it. it's advisable to insert this data and relate it to what you have found as a microbial community.

3. the figures 2 and 4 are not clear, and the small writing is not clearly visible. must be bigger.

4. references format 571-572

5. I want to show an supplementary  table with numbers of sequence reads, good quality reads, ASVs, Shannon diversity, Simpson, Chao 1 indices. 

6. Are you able to go more specific for the analysis of the microbial community? You stopped at level Class, it seems a little reductive according to me, if not at Genus level, but at least at Family level. 

Author Response

Comments and Suggestions for Authors

  1. You must insert a map area.

Response: The map of sampling station was plotted and now as Figure 1. The related description was added in the session of methodology, please see lines 105-107 of the modified manuscript.

  1. Why didn't you measure the redox potential?? For this purpose, given the "extreme" environmental conditions, it would have been important to correlate the structure of the microbial community to the different concentrations of oxygen at various depths. You have postulated a correlation between the structure of the microbial community and variation in environmental parameters, if you have measured the redox, but have not reported it. it's advisable to insert this data and relate it to what you have found as a microbial community.

Response: Thanks for the reviewer’s suggestion. The redox potential is very important, but we didn’t measure it during the cruise due to lacking of suitable equipment. We will consider to include this parameter in the future study.

  1. the figures 2 and 4 are not clear, and the small writing is not clearly visible. must be bigger.

Response: The font size in figures 2 and 4 were enlarged and now are clearly visible.

  1. references format 571-572 (Ref: Parada et al., 2016)

Response: The references format has modified, please see lines 593-595 of the modified manuscript.

  1. I want to show a supplementary table with numbers of sequence reads, good quality reads, ASVs, Shannon diversity, Simpson, Chao 1 indices. 

Response: The numbers of original reads, quality reads and ASVs were already shown in the Table 1; and the Shannon diversity index and Evenness were shown in the Figures 2C and 2D, respectively.

  1. Are you able to go more specific for the analysis of the microbial community? You stopped at level Class, it seems a little reductive according to me, if not at Genus level, but at least at Family level. 

Response: Microbial community at Family level figure were plotted as followed. Because there were too many groups at the Family level, and lots of groups only accounting very low proportions. To be more concise, we chose figures at the class level as in the Figure 2.

Reviewer 2 Report

Comments and Suggestions for Authors

The authors studied the community structure and vertical metabolic activity of two different water trenches.  In general I find this study very interesting and the data presented supports most of the conclusions.  I have two major comments:

1) The authors mentioned that Halobacteria is one of the most dominants microbial groups on both trenches but there is no discussion about their putative metabolic role.

2) Most of the figures have to be redone as there are extremely difficult to interpret (especially Figures 2 and 4)

Comments on the Quality of English Language

Some minor editing

Author Response

The authors studied the community structure and vertical metabolic activity of two different water trenches.  In general, I find this study very interesting and the data presented supports most of the conclusions.  I have two major comments:

  • The authors mentioned that Halobacteria is one of the most dominants microbial groups on both trenches but there is no discussion about their putative metabolic role.

Response: Halobacteria as a class of Euryarchaeota are extremely halophilic archaea that can adapt to a wide range of salt concentration (Gaba et al., 2020) and could catalyze the terminal step in the degradation of organic matter in anoxic environments where light was limiting (Oren and Oremland, 2000). This information was added in the text, please see lines 349-352 of the modified manuscript.

Ref:

Gaba, S., Kumari, A., Medema, M., and Kaushik, R. Pan-genome analysis and ancestral state reconstruction of class halobacteria: probability of a new super-order. Sci Rep 2020, 10(1), 21205. doi:10.1038/s41598-020-77723-6.

Oren, A., and Oremland, R.S. Diversity of anaerobic halophilic microorganisms. Proceedings of SPIE 2000, 4137. doi:10.1117/12.411614.

2) Most of the figures have to be redone as there are extremely difficult to interpret (especially Figures 2 and 4)

Response: The font size in figures 2 and 4 were enlarged and now are clearly visible.

Comments on the Quality of English Language Some minor editing

Response: We had carefully checked the English of the manuscript and corrected the typos. Thanks.

Reviewer 3 Report

Comments and Suggestions for Authors

Dear Editor and dear Authors,

Enclosed you will find my comments to the manuscript “Vertical Metabolic Activity and Community Structure of Prokaryotes along Different Water Depths in the Kermadec and Diamantina Trenches, authored by H. Liu and H. Jing.

In this article, the composition and diversity of microorganisms in two trench areas, Kermadec and Diamantina, are studied. In this approach, through molecular biology quantitative PCR and Biolog EcoplateTM microplates culture, not only the diversity but also the associated carbon metabolisms were studied. The idea was to compare the assemblage of both, diversity and metabolism, throughout the water column, and in the area adjacent to the benthic boundary layers of both sites.

I found the article interesting and appealing. In my opinion, there are some points to be clarified before its publication.

1.     One of my main concerns is the quality of some figures. They are poorly defined, and it makes very difficult to follow the discussion. Please, improve the size and resolution of all the figures.

2.     Figure 2 is especially difficult to read, since all the labels and inserts are so small that the taxonomic groups listed cannot be distinguished. This is a shame since it summarizes one of the main goals of the manuscript. Please, make sure the text in figures is legible, not just in this Figure, but in all.

3.     It could be interesting to determine the relationship between other variables and the diversity. I understand in this occasion you could no take other parameters (DOC, POC, etc.) that could help to make a better inference of the results you got, but please consider it in a future work.

4.     In the discussion section, authors claim that “In the two trenches, significantly different community structure between WC and BBL might be attributed to the in situ concentration and availability of organic matters”. Even if this could be an obvious deduction, authors did not evaluate neither of those variables. So, if you have some data, from your own investigation or other’s research, please include it.

Minor comments:

L-148. Please remove the F in “FThe”

The text could be clearer with little modifications in writting,  for example:

L-338-340 “Generally, BBL contained a high concentration of particles resuspended from subsurface sediments [14,34], might explain the significantly higher gene abundance of prokaryotes in the BBL than in the WC.” It could be better to say “Generally, BBL contain a high concentration of particles resuspended from subsurface sediments [14,34], which could explain the significantly higher gene abundance of prokaryotes in BBL than in WC.”

Comments on the Quality of English Language

There are some typos in the manuscript; the text could be improved if carefully reviewed.

Author Response

Dear Editor and dear Authors,

Enclosed you will find my comments to the manuscript “Vertical Metabolic Activity and Community Structure of Prokaryotes along Different Water Depths in the Kermadec and Diamantina Trenches, authored by H. Liu and H. Jing. In this article, the composition and diversity of microorganisms in two trench areas, Kermadec and Diamantina, are studied. In this approach, through molecular biology quantitative PCR and Biolog EcoplateTM microplates culture, not only the diversity but also the associated carbon metabolisms were studied. The idea was to compare the assemblage of both, diversity and metabolism, throughout the water column, and in the area adjacent to the benthic boundary layers of both sites. I found the article interesting and appealing. In my opinion, there are some points to be clarified before its publication.

  1. One of my main concerns is the quality of some figures. They are poorly defined, and it makes very difficult to follow the discussion. Please, improve the size and resolution of all the figures.

Response: The font size in figures 2 and 4 were enlarged and now are clearly visible. All the figure format fit the submission requirements.

  1. Figure 2 is especially difficult to read, since all the labels and inserts are so small that the taxonomic groups listed cannot be distinguished. This is a shame since it summarizes one of the main goals of the manuscript. Please, make sure the text in figures is legible, not just in this Figure, but in all.

Response: The font size in figures 2 and 4 were enlarged and now are clearly visible.

  1. It could be interesting to determine the relationship between other variables and the diversity. I understand in this occasion you could no take other parameters (DOC, POC, etc.) that could help to make a better inference of the results you got, but please consider it in a future work.

Response: Thank you for your suggestion, we agree that the those paraments such as DOC and POC were important to the microbial communities, and should be measured in the future work.

  1. In the discussion section, authors claim that “In the two trenches, significantly different community structure between WC and BBL might be attributed to the in situ concentration and availability of organic matters”. Even if this could be an obvious deduction, authors did not evaluate neither of those variables. So, if you have some data, from your own investigation or other’s research, please include it.

Response: We totally agree with the reviewer. The benthic boundary layer is the part of the water column that is situated near to the sediment surface, where active oceanic biogeochemical cycling occurs. This layer usually contained a high concentration of particles that are resuspended from the subsurface sediment. The availability of organic matter from those suspended particles was a very important factor determining the community structure in both the sediments and in the thin overlaying sediment-water interface. We would measure those the concentration and composition of those organic matters in the future study.

Minor comments:

L-148. Please remove the F in “FThe”

Response: The typo was corrected, please see line 152 of the modified manuscript.

The text could be clearer with little modifications in writing, for example:

L-338-340 “Generally, BBL contained a high concentration of particles resuspended from subsurface sediments [14,34], might explain the significantly higher gene abundance of prokaryotes in the BBL than in the WC.” It could be better to say “Generally, BBL contain a high concentration of particles resuspended from subsurface sediments [14,34], which could explain the significantly higher gene abundance of prokaryotes in BBL than in WC.”

Response: The sentence was modified according to the reviewer’s suggestion, please see lines 342-344 of the modified manuscript.

Comments on the Quality of English Language

There are some typos in the manuscript; the text could be improved if carefully reviewed.

Response: The whole text was gone through carefully, and typos were corrected. Thanks.

Round 2

Reviewer 3 Report

Comments and Suggestions for Authors

I have no further comments. 

Author Response

Thank you.